# The Evaluation on Corrosion Resistance and Dross Formation of Zn–23 wt % Al–0.3 wt % Si–*x* wt % Mg Alloy

**Wangjun Peng [1,2,3], Guangxin Wu [1,2,3,*], Rui Lu [1,2,3], Quanyong Lian [1,2,3] and Jieyu Zhang [1,2,3]**

[1] State Key Laboratory of Advanced Special Steel, Shanghai University, Shanghai 200072, China; pwj525@shu.edu.cn (W.P.); lurui2578@shu.edu.cn (R.L.); lianquanyong@shu.edu.cn (Q.L.); zjy6162@staff.shu.edu.cn (J.Z.)
[2] Shanghai Key Laboratory of Advanced Ferrometallurgy, Shanghai University, Shanghai 200072, China
[3] School of Materials Science and Engineering, Shanghai University, Shanghai 200072, China
* Correspondence: gxwu@shu.edu.cn

**Abstract:** A comparative study of the corrosive resistance and dross formation of 55Al–Zn–1.6Si (wt %) (55AZS) and 23Al–Zn–0.3Si–*x*Mg (wt %) (23AZS–*x*Mg, *x* = 0, 1.5, 3) alloys are performed using immersion corrosion and dross formation test, respectively. The result of immersion corrosion testing shows that corrosive rate of the 23AZS alloy is lower than that of 55AZS alloy in the latter stage of immersion and 23AZS–1.5Mg alloy shows the optimal corrosive resistance compared to other alloys relatively. The result of dross formation test shows that the number of bottom dross particle formed in 23AZS–*x*Mg (*x* = 0, 1.5, 3) alloy is less than that in 55AZS alloy. Moreover, the thermodynamic calculation is performed to reveal the solubility of Fe in the alloys, the result shows the solubility of Fe reduces as a decrease of Al content in the alloy, and the number of dross particle ($Fe_4Al_{13}$ and $\tau_6$ ($Al_9Fe_2Si_2$) phase) generated in 23AZS alloy is more than that in 55AZS alloy. In general, 23AZS–1.5Mg alloy has an advantage of less dross and a certain corrosion resistance and it is expected to be applied for the hot stamping process of coating.

**Keywords:** 23Al–Zn-0.3Si–*x*Mg (wt %); immersion test; corrosion product; dross formation

## 1. Introduction

The cost of material suffering corrosion occupies around 4% of the GDP of an industrialized nation's economy, which greatly promotes international research in material corrosion. The stability of a material will depend on the protective properties of the surface film formed, if the material is exposed to specific aggressive media [1]. In such a sense, steel electrochemistry protection by means of coating is widely used in various fields, owing to the Al–Zn coating combining the durability of aluminum and the cathodic protection of zinc, not only for more long-term atmospheric corrosion resistance, but also better coating performance and good resistance to high-temperature oxidation, and it has been widely used in bridges, buildings, automobiles, and other industries. Some of the notable Al–Zn commercial alloys are Galvalume®(1972) [2], Galfan®(1980) [3], Lavegal®(1985) [4], etc. Galvalume, as a high Al content coating, is widely used and has received extensive attention. However, it will experience a severe reaction of iron dissolution in a zinc pot at a high temperature of 600 °C when the steel plate contacts the zinc bath. The dissolution rate of iron will increase from 0.8 to 1.8 g/(m²·s) as the content of Al rises from 20 to 30 wt % and the dissolution rate of steel is 2.3 g/(m²·s) when the content of Al rises to 55 wt % [5]. $Fe_2Al_5$, $\tau_5$ ($Al_8Fe_2Si$), and $\tau_6$ ($Al_5FeSi$) phases will be generated and form in the top, floating, and bottom dross in the high Al content condition, once the iron content exceeds its solubility in the zinc bath [6]. The floating dross in the molten bath

mainly includes oxide such as zinc oxide and alumina. The phase relationship in a 55 wt % Al–Zn–Si bath is relatively more complicated than that in the zinc molten bath due to the addition of Al and Si elements in the bath. Many studies report that the bottom dross in the 55 wt % Al–Zn–Si melt contains Fe–Al and Fe–Al–Si intermetallics [7].

One of the main problems in the galvanizing steel industry is the formation of the dross phase on the surface of the rolls, at the bottom of the pot and frequent stoppage of lines. The formation of bottom dross not only wastes the resource and energy, but also impairs the quality of the coating. In addition, salvaging the dross at regular intervals would cause a loss in efficiency. Dross can usually be removed by applying an external magnetic field or a specific device in the process of galvanizing [8–10], but these methods only remove the top or floating dross, it is difficult to deal with the hard and heavy bottom dross [11]. The routine reasons for the formation of bottom dross during the galvanizing process are the high content of Al and the supersaturated iron in zinc pot. On one hand, a decrease of Al content in the zinc bath can significantly reduce the driving force of the Fe–Al reaction [12], reducing the number of bottom dross particles. One the other hand, the density of the bottom dross (Fe–Al and Fe–Al–Si phase) is smaller than the zinc bath if the Al content reduces enough in the zinc bath. The density of the nominal $Fe_4Al_{13}$ ($3.85 \ \mathrm{g/cm^3}$) phase is larger than that of the 55AZS liquid ($3.327 \ \mathrm{g/cm^3}$) and smaller than that of 23AZS liquid ($4.5 \ \mathrm{g/cm^3}$) [13], and the $Fe_4Al_{13}$ phase might act as top or floating dross in the 23AZS bath, thereby reducing the harm to bottom dross. However, it also brings a series of negative effects with the reduction of Al, such as a reduction in corrosion resistance, directly impacting on the product performance.

Zn–Al alloy experiences a transformation of the eutectoid composition at 22.3 wt % Al and the eutectoid structure consists of a dendrite $\alpha$-Al phase and an $\alpha$-Al + η-Zn phase which fills in the interdendritic to the binary area, according phase diagram of Zn and Al [14]. This new eutectoid structure is based on a coating developed by the Teck-Comico Company (Canada), its chemical composition is 23 wt % Al–Zn–0.3 wt % Si, and the coating microstructure consists of internal and external layers: the inner layer of very thin Zn–Fe–Al–Si quaternary alloy and an outer layer of fine eutectoid structure and a coarse eutectic structure. The outer layer of the 23Al–Zn–0.3Si (wt %) coating has good deformation, and no crack is produced owing to a kind of superplastic alloy of the 23Al–Zn–0.3Si (wt %) coating [15]. The metallography of zinc dross in the 23Al–Zn–0.3Si (wt %) coating has been investigated by Joshi [16], and the results show that the $\tau_5$ phase is found in the top of the alloy ingot and with a sublattice model of $Al_{0.66}Fe_{0.19}Si_{0.05}(Al, Si)_{0.1}$.

It is deduced that the number of dross particles will reduce with a decrease of Al content, meanwhile, the corrosive resistance of 23AZS may be deteriorating. The salt immersion test is used to evaluate the corrosive resistance of Zn–Al alloy and the results show the corrosive resistance of Zn–Al alloy will deteriorate until the Al content rises to 23 wt % [17]. Many attempts to improve the corrosive resistance of Zn–Al coatings through adding Mg have been carried out for years in the industry. Coating alloys, like ZAM®(Zn + 6 wt % Al + 3 wt % Mg) [18], SuperDyma®(Zn + 11 wt % Al + 3 wt % Mg + 0.2 wt % Si) [19], and Magnelis®(Zn + 3.5 wt % Al + 3 wt % Mg), have been commonly accepted since the late 1990s. Enhancement of corrosive resistance due to the presence of Mg in zinc-based coatings is well established and a good review is reported in [20]. Several reports [21,22] about the effect of Mg have been published and the results show that Mg is a corrosion-active metal and easily corrodes in water. However, small additions of Mg in Zn alloys (up to 12%) can reduce the corrosion rate of Zn alloy, especially in the case of atmospheric corrosion (e.g., building application). For instance, Tanaka et al. [23] added Mg to the Zn–Al alloy bath and investigated the behavior of the Zn–6 wt % Al–Mg coating, and their results show a beneficial effect of Mg in prevention of corrosion. In the study of Zhu [24], the neutral salt spray and electrochemical test are performed to evaluate the corrosive resistance of the 23Al–Zn–0.3Si (wt %) coating, and the results show that Mg can improve the corrosive resistance and the formability of 23Al–Zn–0.3Si (wt %) coating.

Several reviews of the dross phase and corrosive resistance of 23AZS are reported, while the research on dross phases and corrosion resistance of 23AZS–*x*Mg alloy is still insufficient. In this paper,

the effects of Mg on the microstructure, corrosive resistance, and dross formation of 23AZS are studied by comparing with the different Mg-containing 23AZS and 55AZS alloys. The aim of this work is to develop a novel 23AZS–Mg alloy coating for high-strength steel.

## 2. Experimental

### 2.1. Material

A medium frequency induction furnace was used to smelt the required 23AZS, 23AZS–1.5Mg, 23AZS–3Mg, and 55AZS alloys, and the process of melting was protected by argon gas. The mass of zinc and magnesium were 2.5% and 10% more than the theoretical value in the alloys, respectively, and the rest of the elements were added in accordance with the theoretical value, considering the melting loss. The melted alloy was cast in a copper mold after maintaining at a temperature of 700 °C for 15 min. The actual element compositions of the 55AZS and 23AZS–$x$Mg ($x$ = 0, 1.5, 3) alloys detected by inductively coupled plasma mass spectrometry (ICP-MS, PERKINE 7300DV, PerkinElmer, Boston, MA, USA) are shown in Table 1.

**Table 1.** Chemical composition of the alloys (wt %).

| Alloy | Al | Zn | Si | Mg | La | Fe |
|---|---|---|---|---|---|---|
| 23AZS | 22.80 | bal | 0.38 | – | – | – |
| 23AZS–1.5Mg | 23.20 | bal | 0.34 | 1.58 | – | – |
| 23AZS–3Mg | 23.10 | bal | 0.35 | 2.87 | – | – |
| 55AZS | 52.697 | 45.7 | 1.31 | – | 0.2 | 0.093 |

### 2.2. Experimental Processes

#### 2.2.1. Immersion Test

An immersion test was conducted under the criteria established by GBT19746-2005 standard [25] in this paper. Test specimens with a dimension of 50 mm × 20 mm × 2 mm were cut from the ingot and mechanically ground with 180 to 2000 mesh SiC papers in order, and then these specimens were cleaned in pure alcohol. The procedures above were used to remove oil and other organic contaminants and oxide layers formed during material storage and handling and to build natural oxide layers under controlled conditions. Then, specimens were weighed on an electronic balance with a precision of ±0.1 mg. Next, the specimens were immersed in a natural sodium chloride solution with a salinity of about 36 g/L and located at 20 mm under the liquid level. Finally, the immersion device was moved to a thermostat water bath with a temperature of 30 °C. The schematic diagram of immersion device in shown in Figure 1a. All of the corrosion products on the samples were removed in chromic acid solution (20 g $CrO_3$ + 50 mL $H_3PO_4$ + 1000 mL $H_2O$) at 80 °C following the GBT 16545-1996 standard [26]. The mass loss test was carried out on two samples simultaneously and the result was the average value of the experiment.

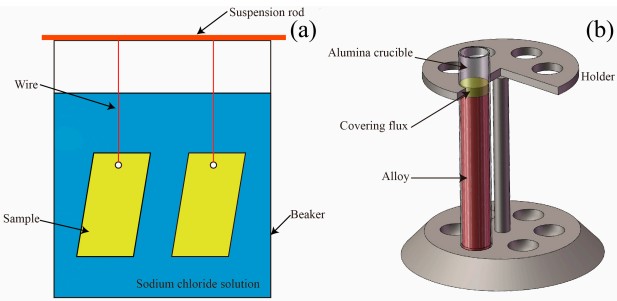

**Figure 1.** Schematic diagram of the experiment: (**a**) Immersion corrosion test [25], and (**b**) dross formation.

### 2.2.2. Dross Generation Test

Covering flux (40 wt % $MgCl_2$, 50 wt % KCl, 5 wt % $BaCl_2$, and 5 wt % $CaF_2$) was melted in an alumina crucible using an electrically-heated crucible furnace (Y-Feng electorical Furnace Co., Ltd, Shanghai, China) to prevent the molten alloys from oxidizing. Then, 23AZS, 23AZS–1.5Mg, 23AZS–3Mg, and 55AZS alloys were added into the crucible and maintained at a temperature of 700 °C for 10 h to ensure the dissolution of raw materials and the homogeneity of bath, respectively. Next, the temperature was decreased to 600 °C and then the pure iron bulk (99.99 wt %, Chuanmao metal materials Co., Ltd, Suzhou, China) was added to the bath with a contact time of 12 h to allow for the precipitation and segregation of dross particles. Finally, the alumina crucible was removed from the furnace and quenched in water to avoid the precipitation of secondary dross particles during solidification. The schematic diagram of the dross formation device is illustrated in Figure 1b.

### 2.3. Experimental Analysis Equipment

The sample was mechanically ground with 360 to 2000 mesh SiC papers in order and then polished using a 0.5 μm diamond paste on a napless polishing cloth. The microstructures of specimen were revealed by 4% nitric acid-alcohol. An examination of the microstructure and element constituent of casting sample was carried out using scanning electron microscopy (SU-1500, HITACHI, Tokyo, Japan) fitted with an INCA energy dispersive spectrometer (EDS) detector (X-Max 20, OXford, OXford, UK). An 18 kW D/MAX2500V + PC X-ray diffractometer (XRD) (Rigaku, The woodlands, TX, USA) was used to obtain RD patterns of casting and immersion samples, and Jade 6.0 software was used to identity crystalline phases by using the JCPDs patterns which have associated the chemical formula of the crystalline phase. The diffractometer scanned from 20° to 90° (2θ) in 4° per minute, using strictly monochromatized Cu Kα-radiation. A camera (5D3, Canon, Tokyo, Japan) was used to catch the morphology of the immersion sample.

## 3. Results and Discussion

### 3.1. Microstructure of the Casting Sample

The phases in Figure 2 are identified by the results of spectral point analysis preliminaries. The dark-gray phase surrounded by the gray phase in Figure 2a is determined to be the Al-rich phase and the atomic ratio of the gray phase is 1:1. The bright-gray phase located at the dendrites of the Al-rich phase is the Zn phase, and some strip-block style phase distributed in the Zn phase is Al, namely, a binary eutectic structure of Zn–Al is formed. The black-block phase embedded in the light-gray phase is Si.

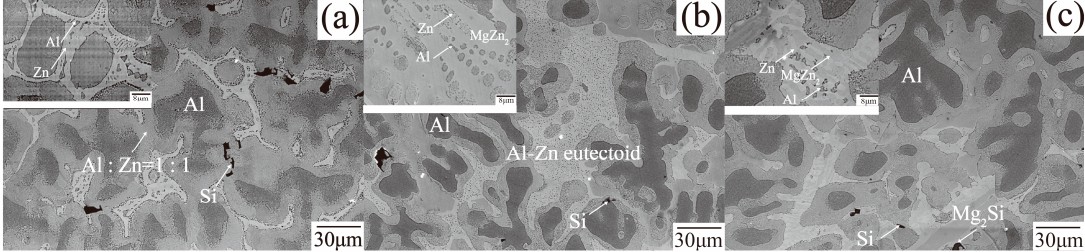

**Figure 2.** The microstructure of alloy: (**a**) 23AZS, (**b**) 23AZS–1.5Mg, and (**c**) 23AZS–3Mg.

The solidification of Al–Zn–Si coating has been reported in the related literature [27–29]. However, there are still some paradoxes of precipitation: for example, the precipitation order of Al–Zn–Si alloy below the melting point. Pandat software (8.5) [30] with the Pan-Al thermodynamic database [31] is used to calculate the phase diagram of 23AlZn–23AlZn1Si (wt %) to obtain phase transitions of the 23AZS alloy below the melting point and it is presented in Figure 3. It is observed that the solidification starts at 490 °C forming primary α-Al (L → α-Al). Then, the reaction of the binary eutectic (L → α-Al

+ Si) begins at 443 °C producing α-Al and Si phases. Next, the crystallization reaction of α-Al occurs at 402 °C. Finally, the reaction of binary eutectoid (Al (Fcc) → Al (Fcc) + Zn (Hcp)) occurs at 277.5 °C, which is consistent with the gray phase in Figure 2a. Therefore, the gray phase is confirmed as the binary eutectoid structure of Al–Zn.

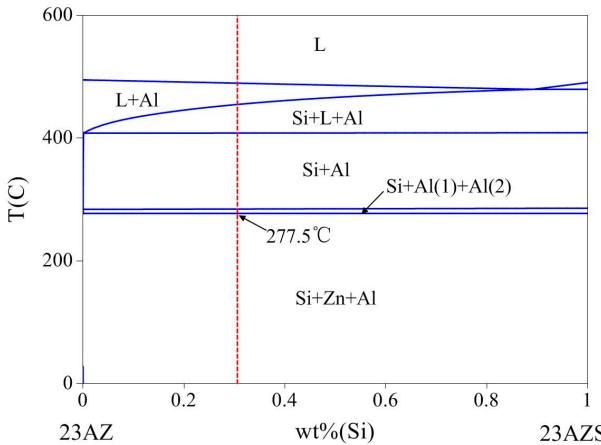

**Figure 3.** Vertical section phase diagram of 23AlZn and 23AlZn1Si (wt %).

The microstructure of 23AZS–1.5Mg alloy is displayed in Figure 2b. A large area of the $MgZn_2$ phase is found in the margin of theAl–Zn eutectoid structure, a light gray $Mg_2Zn_{11}$ phase exists in the dendrite gap, and these phases form the binary eutectic structure of Al and $MgZn_2$ and the ternary structure of $Mg_2Zn_{11}$, Zn, and Al [31]. The main phase composition of 23AZS–3Mg alloy is similar to that of 23AZS–1.5Mg. The bright-gray and black-bulk phases are $Mg_2Zn_{11}$ and $Mg_2Si$, respectively [32]. Al, $Mg_2Zn_{11}$, and Zn phases constitute a ternary eutectic structure, and the black petal-like phase in Figure 2b matches the petal-like phase studied by Chen et al. [27]. The results show that the black petal-like phase is Si, forming an Al–Si binary eutectic structure with the surrounding Al. Meanwhile, this result is consistent with the previous results of our group [33,34].

The phase diagram of Zn23Al0.3Si–Zn23Al0.3Si5Mg (wt %) is used to obtain the influence of Mg on the phase transition of the 23AZS alloy (as shown in Figure 4). It should be noted that the phase diagram shows a process of equilibrium, while the real case is a process of non-equilibrium in our experiment. The result indicates that 23AZS–1.5Mg and 23AZS–3Mg alloys both consist of Si, Al, and $Mg_2Zn_{11}$ phases in the process of equilibrium, however, the $MgZn_2$ phase may appear in the 23AZS–1.5Mg alloy, and $MgZn_2$ and $Mg_2Si$ phases may be found in the 23AZS–3Mg alloy in the process of non-equilibrium, which is in agreement with the results in Figure 2c.

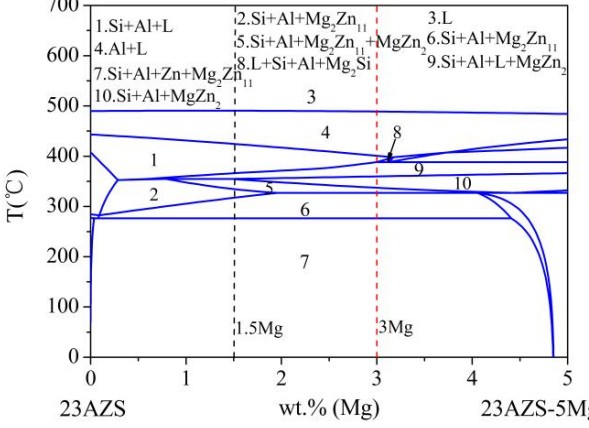

**Figure 4.** Vertical section phase diagram of 23AlZn0.3Si and 23AlZn0.3Si5Mg (wt %).

X-ray diffraction is used to further determine the phase composition of 23AZS–*x*Mg (*x* = 0, 1.5, 3) and it is shown in Figure 5. It can be proved that 23AZS alloy is composed of Zn and Al phases, and 23AZS alloy consists of Zn, Al, $Mg_2Zn_{11}$, and $MgZn_2$ phases after the addition of Mg. The absence of the $Mg_2Si$ phase in the 23AZS–3Mg alloy is caused by the small size and number of $Mg_2Si$ particles (less than 10 μm), as shown in Figure 2c.

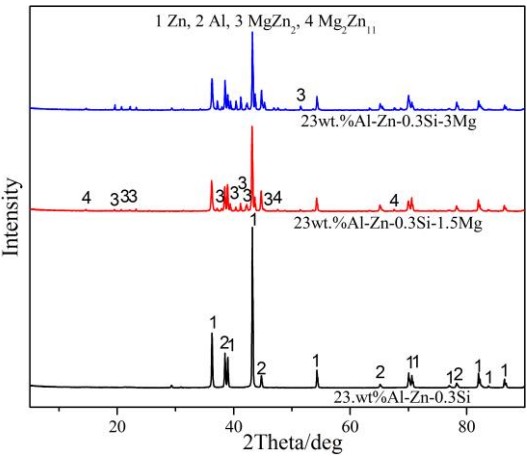

**Figure 5.** X-ray diffraction pattern of 23AZS–*x*Mg (*x* = 0, 1.5, 3).

## 3.2. Microstructure and Phase Composition of Immersion Samples

Corrosion resistance is one of the important indicators for metal material before application, and immersion testing is a method to evaluate the corrosion resistance of different alloy materials. Figure 6 displays the morphology of the 23AZS–*x*Mg (*x* = 0, 1.5, 3) alloy after immersing four and 59 days. All of the samples except the 23AZS–3Mg alloy is completely covered by the corrosion products after immersing for four days, therefore, the corrosion resistance of 23AZS–3Mg alloy is presumed to be higher than that of the other alloys after immersing for four days. Moreover, the color of samples immersing for 59 days is deeper than that of the samples immersing for four days, and all of the samples are completely covered by the corrosion products. In addition, some white point-like corrosion product appear on the surface of the 55AZS alloy.

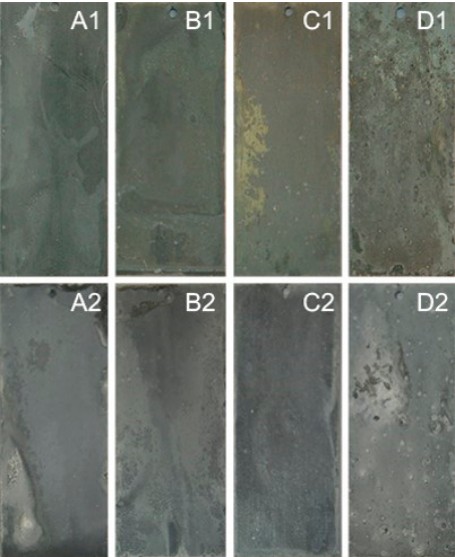

**Figure 6.** The morphology of immersion specimen, A1, B1, C1, D1, A2, B2, C2, and D2 are the 23AZS, 23AZS–1.5Mg, 23AZS–3Mg, and 55AZS after immersing for four and 59 days, respectively.

X-ray diffraction is used to identify the phase composition of corrosion products on the surface of specimen and it is presented in Figure 7. It is indicated that $Zn_5(OH)_8Cl_2 \cdot H_2O$ phase is a primary corrosion product that appears on all specimens, and $Mg_4Al_2(OH)_{12}CO_3 \cdot 3H_2O$ and $Mg_3O(CO_3)_2$ present on the 23AZS specimen after the addition of Mg immersing for four days. However, $Zn_{0.63}Al_{0.37}(OH)_2(CO_3)_{0.185} \cdot xH_2O$ is a corrosion product on 23ASZ and 55AZS specimens after immersing 59 days. Moreover, the diffraction characteristic peak of NaCl in Figure 7 is derived from the residue of the immersion solution, and the Zn, Al, $MgZn_2$, and $Mg_2Zn_{11}$ peaks imply that the X-ray penetrated throughout the whole corrosion product layer. In addition, it is found that $Zn_{0.67}Al_{0.33}(OH)_2(CO_3)_{0.185} \cdot xH_2O$ is added in Figure 7d compared to Figure 7c, and this phase matches the white point-like corrosion product of Figure 6 D2.

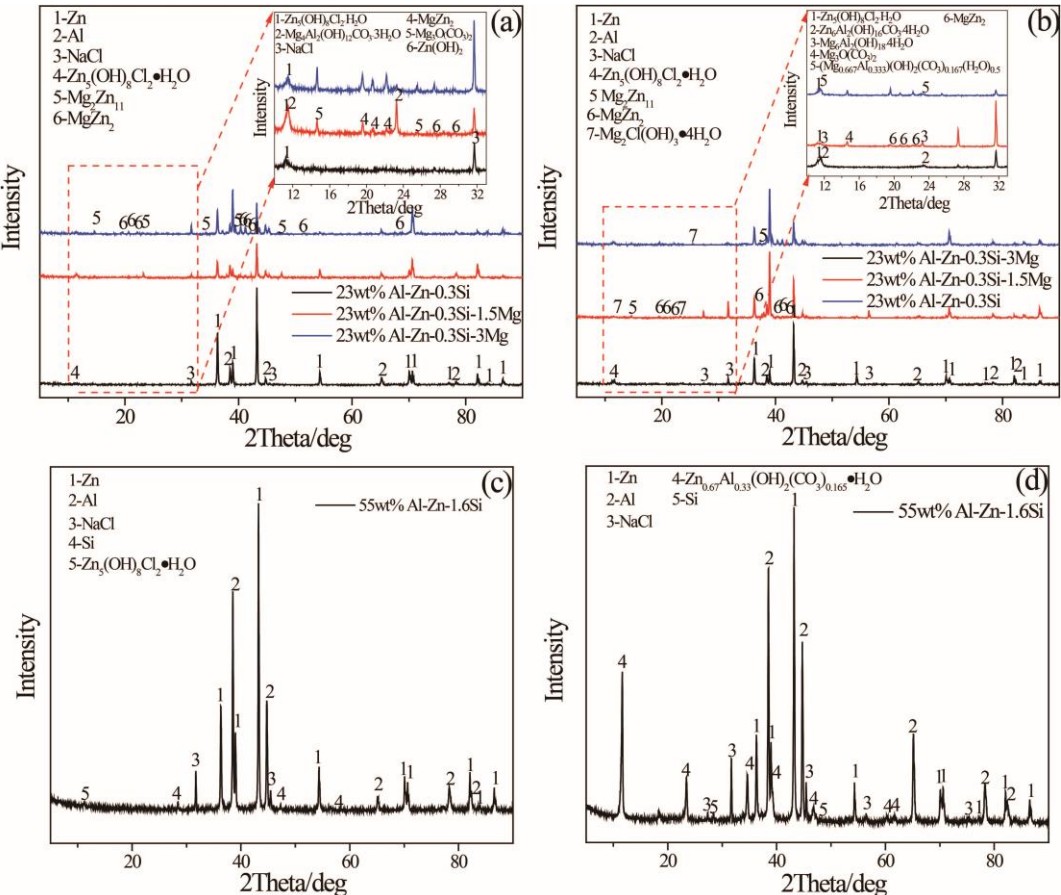

**Figure 7.** X-ray diffraction pattern of immersion specimen: (**a**), (**c**) 23AZS–*x*Mg (*x* = 0, 1.5, 3) and 55AZS alloy immersing for four days, (**b**), and (**d**) 23AZS–*x*Mg (*x* = 0, 1.5, 3) and 55AZS immersing for 59 days.

Mass loss is used to compare the corrosion resistance of different alloys and it is shown in Figure 8. The difference in corrosion rate between 55AZS and 23AZS–*x*Mg (*x* = 0, 1.5, 3) alloys is small at the initial statements of immersion. The corrosion rate of 55AZS (the first slope on the curve) at the beginning is high and the rate decreased (the second slope on the curve in Figure 8) after 30 days of immersion. The corrosion rate decreases due to corrosion inhibition by the formation of $Zn_{0.67}Al_{0.33}(OH)_2 (CO_3)_{0.165} \cdot xH_2O$. After 30 days, the corrosion rate of the alloy 55AZS is very low while the corrosion rate increases for Zn–Al–Mg. A phase transition ($Zn_5(OH)_8Cl_2 \cdot H_2O \rightarrow Zn_{0.67}Al_{0.33}(OH)_2 (CO_3)_{0.165} \cdot xH_2O$) is observed based on the X-ray diffraction result (as shown in Table 2), therefore, it is speculated that the variation of mass loss is attributed to this phase transition.

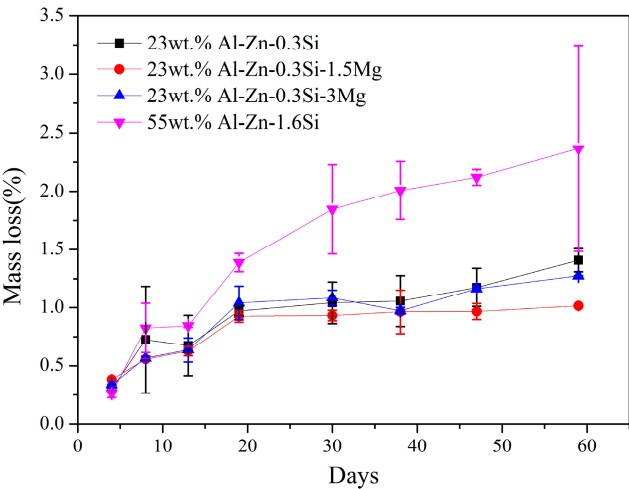

**Figure 8.** Mass loss of immersion specimens.

**Table 2.** The corrosion products of the 55AZS specimen.

| Immersion Days | Detected Phases |
| --- | --- |
| 4 | $Zn_5(OH)_8Cl_2 \cdot H_2O$ |
| 8 | $Zn_5(OH)_8Cl_2 \cdot H_2O$ |
| 13 | $Zn_5(OH)_8Cl_2 \cdot H_2O$ |
| 19 | $Zn_5(OH)_8Cl_2 \cdot H_2O$ |
| 30 | $Zn_5(OH)_8Cl_2 \cdot H_2O$ |
| 38 | $Zn_{0.67}Al_{0.33}(OH)_2(CO_3)_{0.165} \cdot xH_2O$ |
| 47 | $Zn_{0.67}Al_{0.33}(OH)_2(CO_3)_{0.165} \cdot xH_2O$ |
| 59 | $Zn_{0.67}Al_{0.33}(OH)_2(CO_3)_{0.185} \cdot xH_2O$ |

$Zn_5(OH)_8Cl_2 \cdot H_2O$ is observed at anodic sites due to migration of chloride to anodic sites according to Reaction (1):

$$5Zn^{2+} + 2Cl^- + 9H_2O \rightarrow Zn_5(OH)_8Cl_2 \cdot H_2O + 8H^+ \tag{1}$$

$CO_3^{2-}$ is often reported as corrosion products of zinc and hot-dip galvanized steel for the presence of $CO_2$ in air. The formation of $CO_3^{2-}$ depends on the adsorption of carbon dioxide into the surface electrolyte and the formation of $HCO_3^{2-}$ and $CO_3^{2-}$ based on Reactions (2)–(4):

$$CO_2(g) + 2OH^- \leftrightarrow CO_3^{2-} + H_2O \tag{2}$$

$$CO_2(aq) + H_2O \rightarrow HCO_3^- + H^+ \tag{3}$$

$$HCO_3^- \rightarrow CO_3^{2-} + H^+ \tag{4}$$

It should be noted that $Zn_5(OH)_8Cl_2 \cdot H_2O$ undergoes some transition into $Zn_{0.67}Al_{0.33}(OH)_2(CO_3)_{0.185} \cdot xH_2O$ based on Reaction (5):

$$Zn_5(OH)_8Cl_2 \cdot H_2O + Al^{3+} + HCO_3^{2-} + H_2 \rightarrow Zn_{0.67}Al_{0.33}(OH)_2(CO_3)_{0.185} \cdot xH_2O + Cl^- \tag{5}$$

The formation of $Mg_4Al_2(OH)_{12}CO_3 \cdot 3H_2O$ at the cathodic sites is possible based on Reaction (6):

$$4Mg^{2+} + 2Al^{3+} + 12OH^- + CO_3^{2-} + 3H_2O \rightarrow Mg_4Al_2(OH)_{12}CO_3 \cdot 3H_2O \tag{6}$$

Specimens immersing for 30 and 38 days are observed to explain this transition of mass loss for 55AZS and the result is presented in Figure 9. A large surface rupture is observed on the 55AZS specimen immersed for 30 days, while the corrosion product is not found in this feature on the 55AZS specimen immersed for 38 days. On one hand, it is inferred that the stability of $Zn_5(OH)_8Cl_2 \cdot H_2O$ is

lower than that of $Zn_{0.67}Al_{0.33}(OH)_2(CO_3)_{0.165} \cdot xH_2O$ and it can protect the 55AZS alloy from corrosion combining with the results of X-ray diffraction in Figure 7c,d. On the other hand, it is speculated that corrosion products formed during 30 days disappear at 38 days.

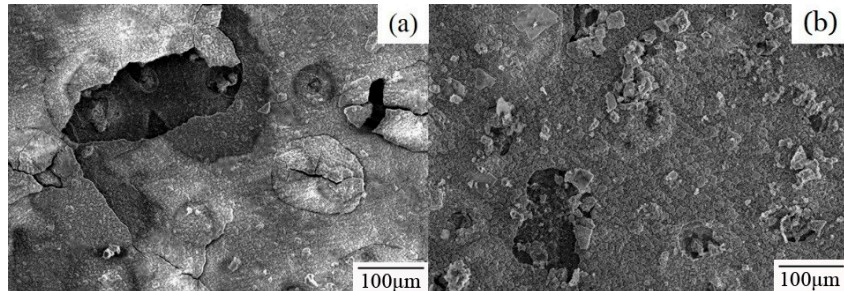

**Figure 9.** The corrosion products microstructure of 55AZS sample exposure for different days: (**a**) 30 days, and (**b**) 38 days.

Figure 10 shows the elemental distribution of 23AZS–3Mg specimen immersed for 47 days. It is observed that the corrosion products are composed of gray and dark-gray phases in Figure 10. Moreover, the distribution of Mg and Cl elements overlaps with the region of the gray phase, therefore, this region is determined to be the place where the reaction of electrochemical occurs ($Mg^{2+} + 2Cl^-$ $\rightarrow MgCl_2$). It is determined that $Mg_2Zn_{11}$, $Mg_2Si$, or $MgZn_2$ phases are the regions where the electrochemical reactions occur [35] combining with the phase composition of 23AZS–3Mg alloy (as shown in Figure 2c). The process of the corrosion reaction is built based on the distribution of element on the 23AZS–3Mg sample (the upper schematic image of Figure 10).

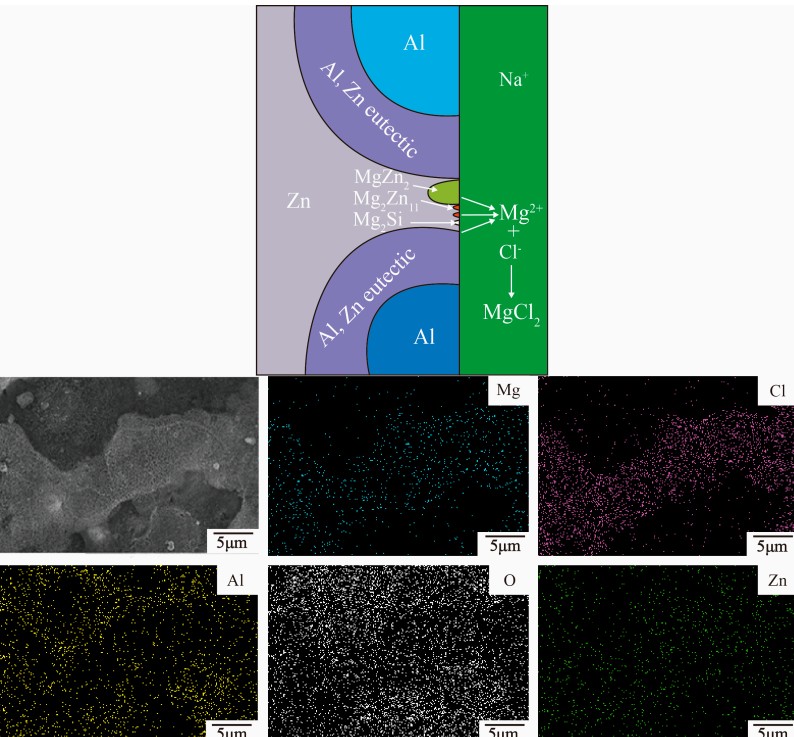

**Figure 10.** The schematic of the corrosion reaction and distribution of element on 23AZS–3Mg immersed for 47 days.

The microstructure of the corrosion products for four group specimens are used to explain the difference in corrosion resistance between 23AZS–$x$Mg ($x$ = 0, 1.5, 3) and 55AZS alloys in the latter stage of immersion (as shown in Figure 11). It is observed that corrosion products on 55AZS specimen appear as visible cracking. Therefore, it also indicates the corrosion resistance of the 23AZS–$x$Mg ($x$ = 0, 1.5, 3) specimen is higher than that of the 55AZS specimen. The corrosion product of the 23AZS coating was studied by Zhu using the neutral salt spray method [24], and the result shows that the corrosion product on Zn–23Al–0.3Si, Zn–23Al–0.3Si–0.5Mg, Zn–23Al–0.3Si–2Mg and Zn–23Al–0.3Si–3.5Mg (wt %) coating mainly consists of $Zn_5(OH)_8Cl_2 \cdot H_2O$. However, $Mg_4Al_2(OH)_{12}CO_3 \cdot 3H_2O$, $Zn_5(OH)_8Cl_2 \cdot H_2O$, and $Mg_3O(CO_3)_2$ phases are found in our immersed specimen. This difference may arise from the different test method and microstructure.

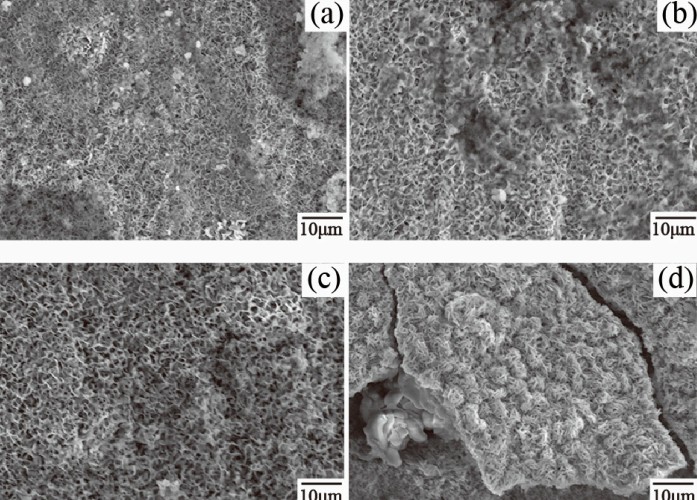

**Figure 11.** The microstructure of corrosion products: (**a**) 23AZS, (**b**) 23AZS–1.5Mg, (**c**) 23AZS–3Mg, and (**d**) 55AZS specimens.

In general, the corrosion resistance of the 23AZS–$x$Mg ($x$ = 0, 1.5, 3) alloy is comparable to that of the 55AZS alloy in the initial stage of immersion, while the corrosion resistance of the 23AZS–$x$Mg ($x$ = 0, 1.5, 3) alloy is higher than that of the 55AZS alloy in the latter stage of immersion, and the 23AZS–1.5Mg alloy shows the best corrosion resistance.

### 3.3. Microstructure and Phase Composition of Dross

The microstructure of bottom dross in 23AZS–$x$Mg ($x$ = 0, 3) and 55AZS alloys are presented in Figure 12. It is speculated that the dross phases are $\tau_5$ ($Al_8Fe_2Si$), $\tau_6$ ($Al_9Fe_2Si_2$), and $Fe_4Al_{13}$ according to the atom ratio of dross phase from the result of EDS point-scanning. The $Fe_4Al_{13}$ phase is a bar-shaped phase and found in 55AZS and 23AZS alloys, the $\tau_6$ phase is needle-like shaped and appears in the 23AZS–1.5Mg alloy, and $\tau_5$ and $Mg_2Si$ phases re polygon-shaped and present in the 23AZS–3Mg alloy. In the results of Liu et al. [36], the $Mg_2Si$ phase is found in the dendrites of the 55Al–Zn–2Mg (wt %) coating and the shape of this phase is similar to our result. It can be concluded that the number of bottom dross particles in 55AZS alloy is obviously more than that in the 23AZS alloy from Figure 12.

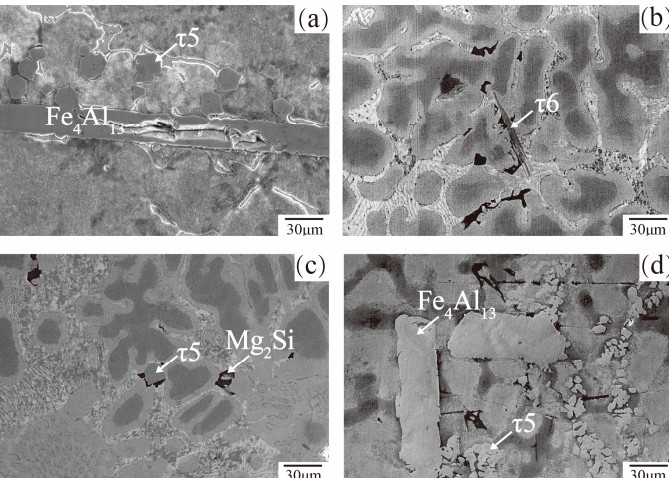

**Figure 12.** The microstructure of bottom dross: (**a**) 23AZS, (**b**) 23AZS–1.5Mg, (**c**) 23AZS–3Mg, and (**d**) 55AZS.

Then, a transmission electron microscope (TEM, JEM-2010F, JEOL, Tokyo, Japan) is used to obtain the diffraction pattern of the dross phase, which further determines the polygon $\tau_5$ phase in the 55AZS alloy. The twin-jet electropolishing method cannot guarantee obtaining the required transmission sample due to the non-uniform distribution of the polygonal $\tau_5$ phase and its size (about 10 μm), therefore, a dual-beam focused ion beam (FIB, 600i, FEI CO., Ltd, Prague, Czech Republic) is used to prepare the transmission sample. The preparation of the transmissive film sample is shown in Figure 13 as follows (Figure 13a): Finding a region to be processed into TEM film in the sample, and using a FIB to rough cut, (Figure 13b); using an Easylift nano-manipulator to cut off the sample and extracted it, (Figure 13c) Sample is moved in high precision and transmitted it to the TEM copper target (Figure 13d); and using a FIB for the preparation of an ultra-thin sample.

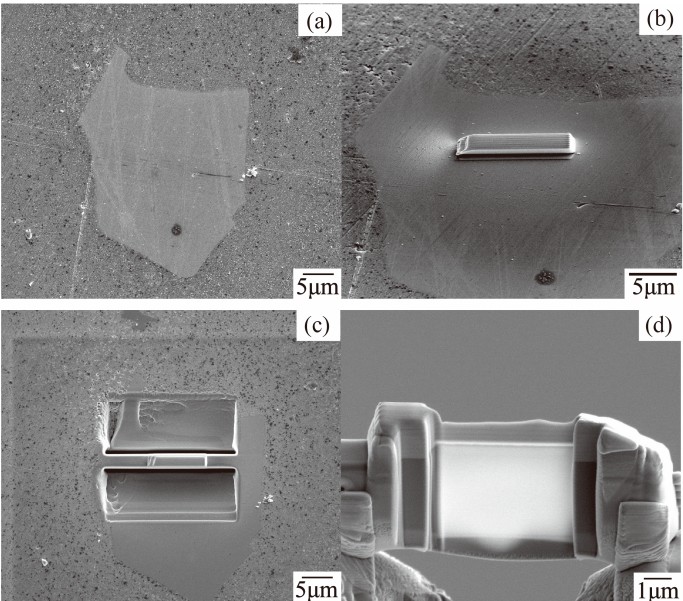

**Figure 13.** SEM images of a TEM film obtained using FIB: (**a**) Found a region, (**b**) Rough cutted, (**c**) Cutted off the sample and extracted it, (**d**) Transmitted it to the TEM copper target and using a FIB for the preparation of an ultra-thin sample.

EDS point-scanning is used to confirm the elements of the transmissive sample and it is presented in Figure 14, showing cross-sectional TEM images of a representative microstructure of the polygonal

$\tau_5$ phase in the 55AZS alloy. It can be observed from the spectrum of (a) and (b) that this phase contains Fe, Zn, Al, Si, and Cu elements. The existence of a Cu peak on the EDS chart is caused by the generation of a characteristic X-ray from the Cu support stand carrying the specimens [37].

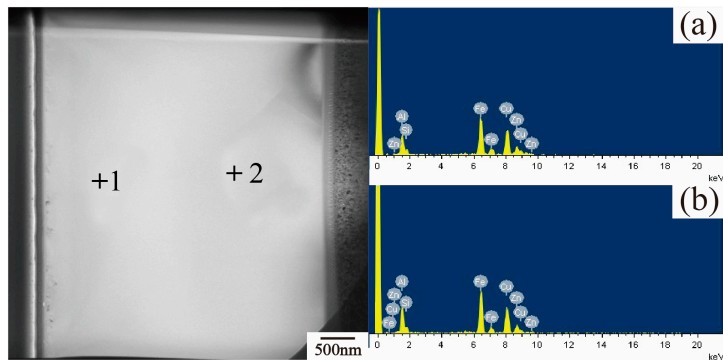

**Figure 14.** TEM bright field image of Fe-Al-Si phase: (**a**) EDS point result of 1, (**b**) EDS point result of 2.

Then, this phase is analyzed by a selected-area electron diffraction pattern to determine the phase of the TEM film sample, as shown in Figure 15. The d-spacing of Figure 16a can be identified with (3 2 6) and (3 1 7) reflections of $\tau_5$, respectively, and the HRTEM (Figure 16b) result shows the d-spacing is 0.8055 nm. The lattice constant of $\tau_5$ ($Al_8Fe_2Si$) for the electron diffraction pattern is obtained from the database 41-0894 of the ICDD. Then, the polygon phase is determined as the $\tau_5$ phase by comparing with the interplanar spacing in the database.

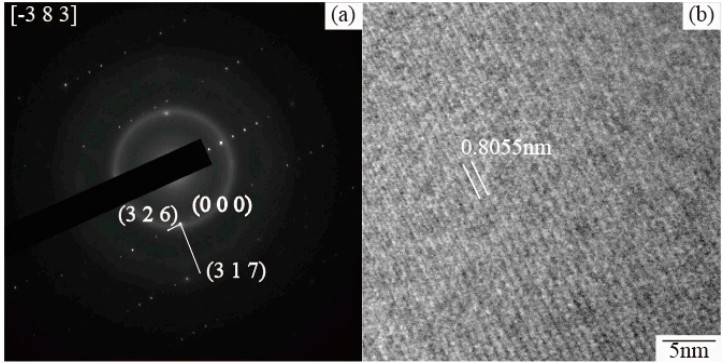

**Figure 15.** (**a**) Diffraction pattern of Fe–Al–Si phase and (**b**) the corresponding HRTEM image taken from area Fe–Al–Si phase.

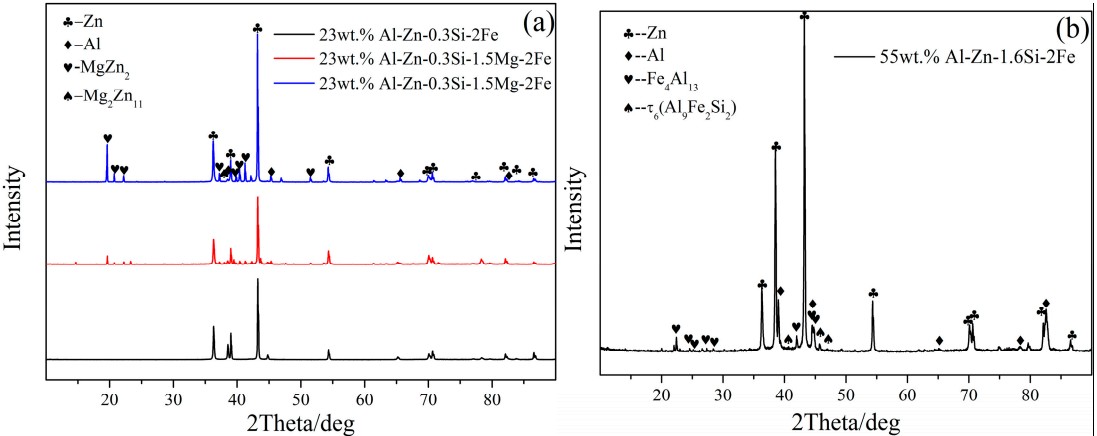

**Figure 16.** X-ray diffraction result of the bottom dros: (**a**) 23AZS–*x*Mg–2Fe, (**b**) 55AZS–2Fe.

Furthermore, X-ray diffraction is used to determine other phases in four group alloys. Figure 16 shows the results of X-ray diffraction at the bottom position in four group alloys. It is shown that the 55AZS alloy contains Zn, Al, $Fe_4Al_{13}$, and $\tau_6$ ($Al_9Fe_2Si_2$) phases, the 23AZS alloy only consists of Zn and Al phases, and both 23AZS–1.5Mg and 23AZS–3Mg alloys are comprised of Zn, Al, $MgZn_2$, and $Mg_2Zn_{11}$ phases. The absence of $\tau_5$ phase in 23AZS–$x$Mg ($x$ = 0, 1.5, 3) and 55AZS alloys caused by non-uniform distribution of the $\tau_5$ phase (as shown in Figure 12).

Next, the distribution of the dross phase is measured with ten images of the microstructure of dross captured for each sample and is shown in Table 3. 55AZS and 23AZS alloys contain $\tau_5$, $\tau_6$, and $Fe_4Al_{13}$ phases, and the 23AZS–1.5Mg alloy comprises $\tau_5$, $\tau_6$, $Fe_4Al_{13}$, $Mg_2Si$, and 'x' phases, the 'x' phase contains Al, Zn, Si, and Mg elements and has an atomic ratio of 69:21:5:5 (as shown in Figure 17). It should be noted that a new tetragonal phase is also observed during the laser treatment of the Zn–Al–Mg–Si coating by Chen et al. [38], and this phase is also composed of Al, Zn, Si, and Mg elements while this phase has an atomic ratio of 16:29:26:28. Thus, the 'x' phase may be a new quaternary phase and we will devote more time to investigate this phase in the future.

**Table 3.** The distribution of the dross phase.

| Alloy | Upper | Middle | Bottom |
|---|---|---|---|
| 55AZS | $\tau_6$ | $\tau_6$ | $\tau_6$, $Fe_4Al_{13}$, $\tau_5$ |
| 23AZS | $\tau_5$, $Fe_4Al_{13}$ | $\tau_5$, $\tau_6$, $Fe_4Al_{13}$ | $\tau_5$, $Fe_4Al_{13}$ |
| 23AZS–1.5Mg | $Mg_2Si$, $\tau_6$, $\tau_5$, x | $Mg_2Si$, $\tau_6$, $\tau_5$, x | $Mg_2Si$, $\tau_6$, $\tau_5$, x |
| 23AZS–3Mg | $\tau_5$, $Mg_2Si$ | $Mg_2Si$, $\tau_5$, $Fe_4Al_{13}$ | $Mg_2Si$, $\tau_5$ |

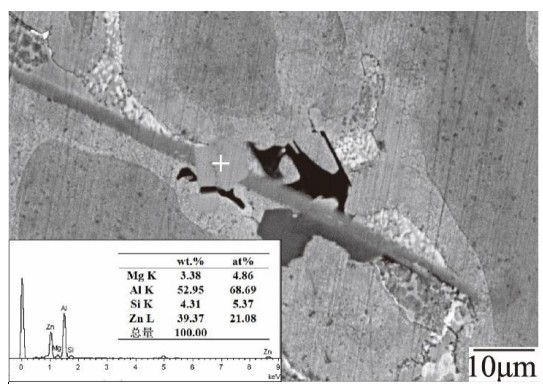

**Figure 17.** EDS result of the 'X' phase.

The relationship between the solubility of aluminum and iron in zinc bath is calculated at the experimental temperature 600 °C using Pandat 8.0 software [30] and with the Pan-Al thermodynamic database [31] to obtain the quantificational results of the dross phase. The dissolution of iron is proportional to the amount of dross in the zinc pot (as shown in Figure 18). The solubility values at the experimental temperature 600 °C are extracted from the solubility curves and shown in Table 4. The solubility of iron increases from 0.002 to 0.004 wt % as Mg content rises from 0 to 3 wt %, and the solubility of iron for the 55AZS alloy (0.01 wt %) is larger than that of the 23AZS–$x$Mg alloy ($x$ = 0, 1.5, 3) (0.002–0.004 wt %), which can reflect the number dross particles of the 23AZS–$x$Mg alloy ($x$ = 0, 1.5, 3) is less than that of the 55AZS alloy. Next, the iron solubility curve of the 23ASZ alloy with different Si content is also calculated (as shown in Figure 19). On the one hand, it indicates the solubility of iron increases with the Si content increasing. On the other hand, it also shows that more dross phases will be precipitated from the 55AlZn alloy and the total number of dross phase particles in the 23AlZn alloy will decrease.

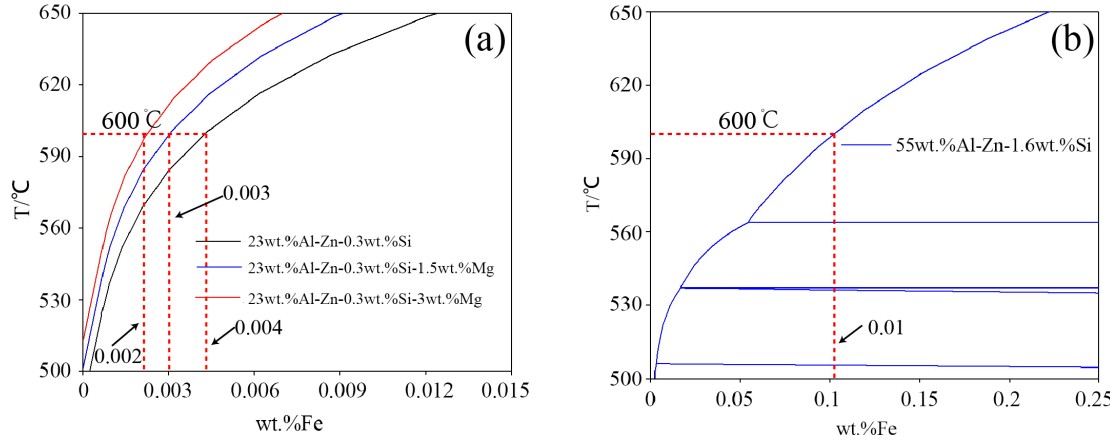

**Figure 18.** The solubility curves of Fe in different alloys: (**a**) 23AZS–*x*Mg ally, (**b**) 55AZS alloy.

**Table 4.** The solubility of Fe in different alloys.

| Alloy name | Solubility/wt % | Alloy | Solubility/wt % |
|---|---|---|---|
| 23AZS | 0.002 | 23ASZ–3Mg | 0.004 |
| 23ASZ–1.5Mg | 0.003 | 55AZS | 0.01 |

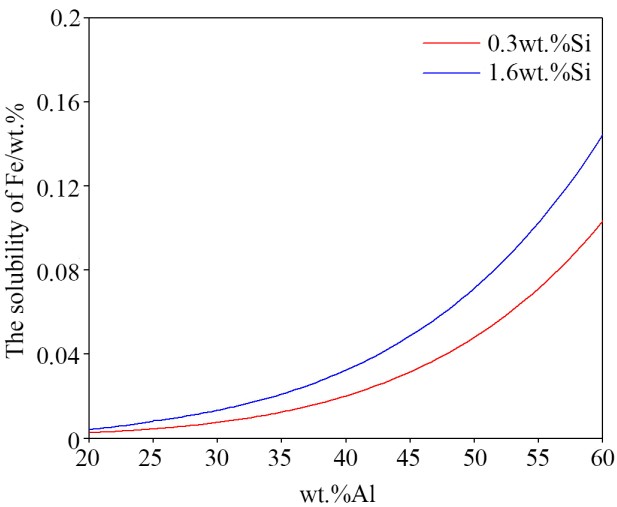

**Figure 19.** The solubility of Fe in bath with different Al content.

The phase fractions of $Fe_4Al_{13}$ and $\tau_6$ phases which are the main dross phases in four group alloys are calculated and presented in Figure 20. The phase fractions of $Fe_4Al_{13}$ and $\tau_6$ phases in 55AZS alloy are 0.058 and 0.084, respectively, and the phase fractions of $Fe_4Al_{13}$ and $\tau_6$ phases in the 23AZS alloy are 0.038 and 0.030, respectively. This means that the phase fractions of $Fe_4Al_{13}$ and $\tau_6$ phases in the 23AZS alloy are less than that in the 55AZS alloy, due to the large solubility difference of iron between 55AZS and 23AZS alloys (as shown in Table 4).

The dross composition of 22.3Al–Zn–0.6Si (wt %) at a temperature range of 480–620 °C is studied by Pan et al., and the result shows that the zinc dross formed on surface of 22.3Al–Zn–0.6Si (wt %) at 620 °C is composed of $\tau_2$ and $FeAl_3$ phases [39], while $\tau_5$ and $FeAl_3$ are the main phases formed on 22.3Al–Zn–0.3 Si (wt %) alloy. The possible reason is the different Si content and temperature. Otherwise, the corrosion potentials of dross phase have been reported by many researchers, and the result is shown in Table 5. It could be inferred that corrosion potentials of $Mg_2Si$ $MgZn_2$, Al, and Al(Zn) phases are lower than the Fe phase. However, the Si, Fe–Al, and Fe–Al–Si phases are opposite. Therefore, some dross phase, such as $Mg_2Si$ $MgZn_2$, Al, and Al(Zn), can provide cathodic

protection, and other phases, such as Fe–Al and Fe–Al–Si intermetallic phases, will deteriorate the electrochemical protection for steel.

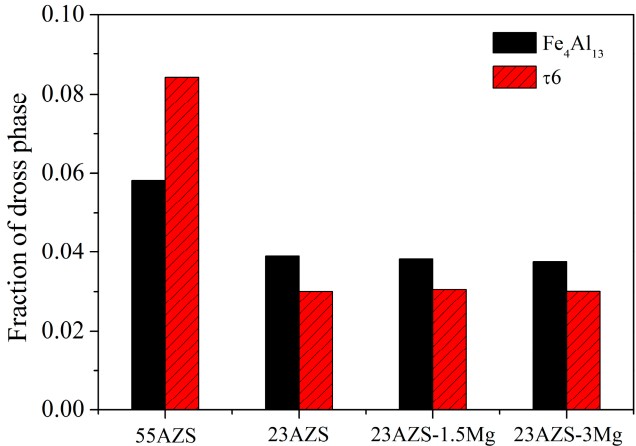

**Figure 20.** The fraction of $Fe_4Al_{13}$ and $Al_9Fe_2Si_2$ phases.

**Table 5.** Corrosion potentials of dross phase in related references [40–42].

| Phase | Composition | *E*/mV vs. SCE |
|:-----:|:-----------:|:--------------:|
| 1 | Al | −823 [40] |
| 2 | Al(Zn) | −920 [40] |
| 3 | Si | −441 [40] |
| 4 | $Al_xFe_ySi$ ($\tau_5$) | −470 [40] |
| 5 | $Mg_2Si$ | −1538 [40] |
| 6 | $MgZn_2$ | −1029 [40] |
| 7 | $FeAl_3$ | −539 [41] |
| 8 | Fe | −698 [42] |

In summary, 23AZS–1.5Mg alloy has better corrosion resistance and lower dross phase content compared with 55AZS alloy, which is supposed to improve the number of $Mg_2Si$ and $MgZn_2$ particles and reduce the number of Fe–Al and Fe–Al–Si intermetallic particles in the coating. Meanwhile, it should be evaluated that it has a superior formability according to its eutectoid structure. It is supposed to possibly replace the existing galvanized and 55Al–Zn alloy coating.

## 4. Conclusions

In this work, the natural corrosion and dross formation of 23AZS–$x$Mg ($x$ = 0, 1.5, 3) and 55AZS alloys were investigated. The following conclusions can be drawn:

- The 23AZS and 55AZS alloys consist of Al, Zn, and Si phases, and the 23AZS–1.5Mg and 23AZS–3Mg alloys consist of Al, Zn, $Mg_2Si$, $MgZn_2$, and $Mg_2Zn_{11}$ phases.
- In general, the corrosive resistance of different alloys are equal in the initial stage, while the corrosive resistance of 23AZS–$x$Mg ($x$ = 0, 1.5, 3) is higher than 55AZS in the latter stage, and the 23AZS–1.5Mg alloy shows the optimal property in this investigation.
- The corrosion rate of 55AZS at the beginning is high, the rate decreased after 30 days of immersion, and the corrosion rate decreased due to corrosion inhibition by the formation of $Zn_{0.67}Al_{0.33}(OH)_2$ $(CO_3)_{0.165} \cdot xH_2O$.
- The bottom dross of 55AZS and 23AZS alloys mainly contain $\tau_6$, $Fe_4Al_{13}$ and $\tau_5$ phases, the solubility of Fe calculated by Pandat will increase with the increase of Al content. Thus, the amount of dross phase ($Fe_4Al_{13}$ and $\tau_6$) in the 23AZS coating is less than that of 55AZS alloy, which is consistent with our experimental results.

**Author Contributions:** Conceptualization, G.W. and J.Z.; Methodology, W.P.; Software, W.P.; Validation, G.W. and J.Z.; Formal Analysis, W.P.; Investigation, W.P.; Resources, R.L.; Data Curation, Q.L.; Writing—Original Draft Preparation, W.P.; Writing—Review and Editing, W.P.; Visualization, W.P.; Supervision, G.W.; Project Administration, G.W.; Funding Acquisition, G.W.

**Funding:** This research was funded by National Natural Science Foundation of China (Grant No. 51674163) and Science and Technology Committee of Shanghai under No. 16ZR1412000.

**Acknowledgments:** The author Wu would like to thanks for Analysis and Test Center of Shanghai University for the support of instrument.

**Conflicts of Interest:** The authors declare no conflict of interest.

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
