# Peer review of "The Evaluation on Corrosion Resistance and Dross Formation of Zn–23 wt % Al–0.3 wt % Si–x wt % Mg Alloy"

_coatings, doi:10.3390/coatings9030199_

Reviewer 1 Report

Review

 The Evaluation on Corrosion Resistance and Dross  Formation of Zn-23 wt.% Al-0.3 wt.% Si-x wt.% Mg  Alloy

1.I did not find any changes in red in new text. 

2. The last comments. I am not specialist in English, but the article is really needs in English editing. Many phrases are badly formulated and unclear.

3. Line 80. “published and the results show that Mg have low density and high electronegativity among”

4. The electronegativity, it is property of one single atom to attract the electron pair in the chemical bond with other atom. According Pauling electronegativity is Mg= 1.31, Fe= 1.83 and Al= 1.61 (Wikipedia).

5. I propose to replace ... Mg is corrosion active metal and easily corrode in water. However small additions of Mg in Zn alloys (up to 12%) decreases corrosion rate of Zn alloy especially in case of atmospheric corrosion (e.g. building application).

6. I don’t understand the value of Figure 9 and corresponding phrase. I advise you to remove the Figure 9.

7. Figure 9, Distribution of species in corrosion products formed on specimens immersed to natural

sodium chloride solution for 59 days

8. Line 228     “The composition of water soluble and corrosion products dissolved in chromic acid solution in  terms of Cl-, Na+, Zn2+, Mg2+, OH-, H+, Al3+ and CO32- to analyze the phase transition during immersion, a picture of the surface chemistry is obtained after immersing days (As shown in Figure 9. The Na element is an active metal and the potential is low than Mg, Al and Zn, and it means the Na+ cannot react with the element in alloys. Moreover, the corrosive production containing Na+ is not observed in immersed alloy according to the result of XRD, therefore, Na+ doesn’t give mass gain in alloy. “  The phrase has to be removed.

10. The corrosion products are formed on the alloy surface and are not in chromic acid. They consist from oxides-hydroxides-carbonates of corresponding metals. They creates corresponding phases but these phases are not the layers as you show in the Figure 9. It is more or less uniformly distributed mixture of the phases in the surface layer in air or in the water. You cannot make any analyses after dissolution of products in chromic acid. The acid treatment is only for determination of mass lost.

11. “Na+ doesn’t give mass gain in alloy.”   You determined mass lost, it is not gain. 

12. Na+ doesn’t disappear and it is contained in corrosion products as Na2CO3 or NaHCO3 and dissolved in water. They can not be found bt XRD.  In chromic acid, these compounds are not stable. Please remove the phrase, it shows low the electrochemistry level. All peoples know that reduction of Na+ to Na or Mg2+ to Mg or Al3+ to Al is impossible in aqueous electrolytes.

13. Line 245 “carbonization”. Reaction 5 is formation of mixed Zn-Al hydroxy- carbonates. Remove “carbonization”.

14. Line 252 “30 days, while the corrosion product is not found this feature on 55AZS specimen immersing 38 days”. Is it mean, corrosion products formed during 30 days disappear at 38 d?

15. Figure 16. I don’t understand why this figure and experiments are needed. What they show? Conclusion?

Author Response

Review1:

The Evaluation on Corrosion Resistance and Dross  Formation of Zn-23 wt.% Al-0.3 wt.% Si-x wt.% Mg  Alloy

1. I did not find any changes in red in new text. 

We are very sorry for our negligence of the changes in red. The article was submitted once before, and the modified part was completely cancelled after resubmitting.

2. The last comments. I am not specialist in English, but the article is really needs in English editing. Many phrases are badly formulated and unclear.

Thanks for your advice. We have revised the whole manuscript carefully, especially for the grammatical errors and sentences.

3. Line 80. “published and the results show that Mg have low density and high electronegativity among coating metal” The electronegativity, it is property of one single atom to attract the electron pair in the chemical bond with other atom. According Pauling electronegativity is Mg= 1.31, Fe= 1.83 and Al= 1.61 (Wikipedia). I propose to replace ... Mg is corrosion active metal and easily corrode in water. However small additions of Mg in Zn alloys (up to 12%) decreases corrosion rate of Zn alloy especially in case of atmospheric corrosion (e.g. building application).

Thanks. The line “Mg have low density and high electronegativity among coating metal” has been replaced by “Mg is corrosion active metal and easily corrode in water. However small additions of Mg in Zn alloys (up to 12%) can reduce the corrosion rate of Zn alloy especially in case of atmospheric corrosion (e.g. building application).". And it was shown in revised manuscript.

4. I don’t understand the value of Figure 9 and corresponding phrase. I advise you to remove the Figure 9.

Thanks for your advice. I have taken your advice and the Figure 9 has been removed.

5. Figure 9, Distribution of species in corrosion products formed on specimens immersed to natural sodium chloride solution for 59 days. Line 228 “The composition of water soluble and corrosion products dissolved in chromic acid solution in  terms of Cl-, Na+, Zn2+, Mg2+, OH-, H+, Al3+ and CO32- to analyze the phase transition during immersion, a picture of the surface chemistry is obtained after immersing days (As shown in Figure 9). The Na element is an active metal and the potential is low than Mg, Al and Zn, and it means the Na+ cannot react with the element in alloys. Moreover, the corrosive production containing Na+ is not observed in immersed alloy according to the result of XRD, therefore, Na+ doesn’t give mass gain in alloy. “The phrase has to be removed.

Thanks. The line “The composition of water soluble and corrosion products dissolved in chromic acid solution in  terms of Cl-, Na+, Zn2+, Mg2+, OH-, H+, Al3+ and CO32- to analyze the phase transition during immersion, a picture of the surface chemistry is obtained after immersing days (As shown in Figure 9). The Na element is an active metal and the potential is low than Mg, Al and Zn, and it means the Na+ cannot react with the element in alloys. Moreover, the corrosive production containing Na+ is not observed in immersed alloy according to the result of XRD, therefore, Na+ doesn’t give mass gain in alloy” has been removed.

6. The corrosion products are formed on the alloy surface and are not in chromic acid. They consist from oxides-hydroxides-carbonates of corresponding metals. They creates corresponding phases but these phases are not the layers as you show in the Figure 9. It is more or less uniformly distributed mixture of the phases in the surface layer in air or in the water. You cannot make any analyses after dissolution of products in chromic acid. The acid treatment is only for determination of mass lost.

“Na+ doesn’t give mass gain in alloy.”   You determined mass lost, it is not gain. 

Na+ doesn’t disappear and it is contained in corrosion products as Na2CO3 or NaHCO3 and dissolved in water. They can not be found bt XRD.  In chromic acid, these compounds are not stable. Please remove the phrase, it shows low the electrochemistry level. All peoples know that reduction of Na+ to Na or Mg2+ to Mg or Al3+ to Al is impossible in aqueous electrolytes.

Thanks. We have been removed this phrase refers to the advice above.

7. Line 245 “carbonization”. Reaction 5 is formation of mixed Zn-Al hydroxy- carbonates. Remove “carbonization”.

Thanks. The Line 245 “carbonization” has been removed and it was shown in revised manuscript.

 8. Line 252 “30 days, while the corrosion product is not found this feature on 55AZS specimen immersing 38 days”. Is it mean, corrosion products formed during 30 days disappear at 38 d?

Thanks. The line “30 days, while the corrosion product is not found this feature on 55AZS specimen immersing 38 days” has been replaced by “On one hand, it is inferred that the stability of Zn5(OH)8Cl2·H2O is lower than that of Zn0.67Al0.33(OH)2(CO3)0.165·xH2O and it can protect 55AZS alloy from corrosion combining with the results of X-ray diffraction in Fig.7c and d. One the other hand, it is speculated corrosion products formed during 30 days disappears at 38 days.”, and it was shown in revised manuscript.

9. Figure 16. I don’t understand why this figure and experiments are needed. What they show? Conclusion?

Thanks. The figure 16 is a selected-area electron diffraction pattern of Fe-Al-Si phase and corresponding HRTEM image taken from area Fe-Al-Si phase. This figure is to determine that the polygonal of Fe-Al-Si phase is τ5 phase. It can only prove that alloys contain τ5 phase, but cannot be sure which phase is τ5 phase based on the EDS and XRD result of dross in Figure 17.

Reviewer 2 Report

The presented paper is a thorough report on corrosion resistance and dross formation of specific hot-dipping alloys. The results and discussion are strictly technical, and may be of interest for engineering purposes. I have several remarks: -

  1. The introduction could be clarified, mainly with motivation of the work - why this specific additive (Mg) was chosen, and not e.g. Si? -

  2. What about the industrial utilization which is only briefly mentioned in the abstract? -

  3. Some of the figures are intelligible, especially the structure photographs with black lettering (as in Figure 2) -

  4. The conclusions could be elaborated, especially in view of initial statements of corrosion protection costs - is the presented solution better?

Author Response

Review2:

The presented paper is a thorough report on corrosion resistance and dross formation of specific hot-dipping alloys. The results and discussion are strictly technical, and may be of interest for engineering purposes. I have several remarks: -

1.The introduction could be clarified, mainly with motivation of the work - why this specific additive (Mg) was chosen, and not e.g. Si?

Thanks. The general method of improving the corrosion resistance of the coating in the industry are adding a certain amount of Mg, such as ZAMÒ, SuperDymaÒ, MagnelisÒ and 55 wt.% AlZnSi-2Mg, and these coatings are some mature and industrialized product. The effect of adding Si is mainly to suppress the thickness of the Fe-Al alloy layer during the process of galvanizing. This paper is optimizing the Mg content of mature 23wt.% Al-Zn-0.3Si coating to improve the corrosion resistance of coating.

2.– What about the industrial utilization which is only briefly mentioned in the abstract?

Thanks. This product (23Al-Zn-0.3Si) is a new coating product developed by Teck-Comico Company (Canada). The advantage of this coating is that it reduces the number of bottom dross while maintaining a certain corrosion resistance, at the same time, it has a good deformation. It has not been promoted globally after the product was developed. The lack of corrosion resistance has led to certain limitations on its application, therefore, this paper hope to add a certain amount of Mg to improve the corrosion resistance of the coating.

3.– Some of the figures are intelligible, especially the structure photographs with black lettering (as in Figure 2)

Thanks. The description of figures have been revised and some figures have been modified, and it was shown in revised manuscript.

4.– The conclusions could be elaborated, especially in view of initial statements of corrosion protection costs - is the presented solution better? 

Thanks. The conclusions have been revised, and “the view of initial statements of corrosion protection costs” have been added. “The difference in corrosion rate between 55AZS and 23AZS-xMg (x=0, 1.5, 3) alloy is small at the initial statements of immersion.”, and it was shown in revised manuscript.

Reviewer 3 Report

1. In this manuscript, the authors report the results of the corrosion test and dross formation test. In the corrosion test, four types of the specimens were tested. As written in the objective, it is important to find a corrosion resistant material by performing a corrosion test for the application.

In introduction, the previous works and the objective of the present work are clearly described. Then experimental explanation and results & discussion are written. One of the problems in this manuscript is that the authors display so many figures and tables which are not directly related to the conclusion of the research. This makes the manuscipt messy, and I expect potential readers will be confused by this .

2. One of the main results should be the mass loss of the specimens (Fig.8). Based on this results, the authors concluded that 23 AZS-1.5Mg has an optimal property. However, the difference among the four specimens were very small (the differences were as large as experimental error). Moreover, unit of mass loss should be % to original mass, as original weights of the four samples should be different.

3. Let me point out that the presentations of the figures in the manuscript are very poor; sometimes it is difficult/impossible to read written words in a graph (e.g. Figs. 15, 18, and 21). In “2. Experimental”, description about HR-TEM is missing. Meaning of Fig. 9 is unclear. No explanation is given for the upper schematic image in Fig. 11. The written English should be moderately revised, e.g. by using an academic English correction service.

4. Although I agree the contents of the results are valuable for readers, significant efforts should be made by the authors before publishing this manuscript. Thus, let me conclude that my suggestion is Reject.

Author Response

Review3:

1. In this manuscript, the authors report the results of the corrosion test and dross formation test. In the corrosion test, four types of the specimens were tested. As written in the objective, it is important to find a corrosion resistant material by performing a corrosion test for the application. In introduction, the previous works and the objective of the present work are clearly described. Then experimental explanation and results & discussion are written.

One of the problems in this manuscript is that the authors display so many figures and tables which are not directly related to the conclusion of the research. This makes the manuscipt messy, and I expect potential readers will be confused by this.

Thanks. Some description of figures and tables have been revised and some figures have been modified, and it was shown in revised manuscript.

2. One of the main results should be the mass loss of the specimens (Fig.8). Based on this results, the authors concluded that 23 AZS-1.5Mg has an optimal property. However, the difference among the four specimens were very small (the differences were as large as experimental error).

Thanks. The mass loss test is carried out on two samples simultaneously and the result is the average value of the experiment, so the experimental results are real and effectively. The microstructure of the alloy is different from that of the coated products. Therefore, the research results of the alloy have a certain reference significance of corrosion resistance for the coating, and we will continue to study the performance of coating products.

3. Moreover, unit of mass loss should be % to original mass, as original weights of the four samples should be different.

Thanks. The unit of mass loss in Figure 8 has been revised by mass percentage, and it was shown in revised manuscript.

4. Let me point out that the presentations of the figures in the manuscript are very poor; sometimes it is difficult/impossible to read written words in a graph (e.g. Figs. 15, 18, and 21). In “2. Experimental”, description about HR-TEM is missing.

Thanks. The quality of figure such as Figure 3, 4, 5, 7, 8,15, 16, 18, 19, and 21 have been improved, and the description of HR-TEM has been revised “Then, this phase is analyzed by a selected-area electron diffraction pattern to determine the phase of TEM film sample and shows in Figure 15. The d-spacing of Figure 16(a) can be identified with (3 2 6) and (3 1 7) reflections of τ5 respectively, and the HRTEM (Figure 16(b)) result shows the d-spacing is 0.8055 nm. The lattice constant of t5 (Al8Fe2Si) for the electron diffraction pattern is obtained from the database 41-0894 of the ICDD. Then, the polygon phase is determined as τ5 phase by comparing with the interplanar spacing in the database.”, and it was shown in revised manuscript.

5. Meaning of Fig. 9 is unclear.

Thanks. Fig. 9 has been removed according to the review above.

6. No explanation is given for the upper schematic image in Fig. 11. The written English should be moderately revised, e.g. by using an academic English correction service.

Thanks for your advice. The explanation for the upper schematic image in Fig. 11 has been added “The process of the corrosion reaction is built based on the distribution of element on 23AZS-3Mg sample (The upper schematic image of Fig. 10).”. We have revised the whole manuscript carefully, especially for the grammatical errors and sentences.

7. Although I agree the contents of the results are valuable for readers, significant efforts should be made by the authors before publishing this manuscript. Thus, let me conclude that my suggestion is Reject.

Thanks very much for your kind work and consideration on publication of our paper. We tried our best to improve the manuscript and made some changes in the manuscript. And we believe that the language is now acceptable to presentation of the work. Once again, thank you very much for your comments and suggestions.

Round  2

Reviewer 1 Report

Dear authors thank you. 

kind regards

Reviewer 2 Report

Dear Authors,

I see that my suggestions were taken into account, and I have no further remarks.

Maybe You could improve the black captions on dark gray photos in figure 2 and 12 - they are hard to read, but that only editorial remark.

Sincerely, 

Reviewer 

Reviewer 3 Report

The authors have modified the manuscript based on the reviewer's comments.I think the contents of this work are interesting for readers. I agree to accept the manuscript with the current form.

This manuscript is a resubmission of an earlier submission. The following is a list of the peer review reports and author responses from that submission.

Round  1

Reviewer 1 Report

The authors present some interesting results; however, the quality of presentation is very low, the paper is hard to read and the are many language issues that makes scientific evaluation very hard.

I would recommend that the authors rewrite and reorganize the paper and resubmit it.

Reviewer 2 Report

Review to the article: The evaluation on corrosion resistance and dross formation of Zn-23 wt.% Al-0.3 wt.% Si-x wt.% Mg  alloy”  Wangjun Peng, Guangxin Wu, Rui Lu, Quanyong Lian, Jieyu Zhang, submitted for publication in METALS, MDPI

1. The article investigates the metallurgy of Zn-Al- Mg Si and Al-Zn alloys, describes the formation dross phases Fe4Al13 and (Al9Fe2Si2), determines the relative corrosion stability in aqueous electrolyte.

2. The main emphasis of the article was on preparation and phase characterization of the intermetallic and mixed alloy phases. Metallography, TEM, SEM-EDS and XRD techniques were employed. The relative corrosion resistance was determined by mass lost measurements

3. Line 28. “ and anode protection of zinc” 

4. “Zn” or “Zn-Mg” alloys coatings perform cathodic protection of the steel, shifting the potential to cathodic direction.

5. line 31.   “Some of noTablele Al-Zn commercial” it is not clear, mistake in English?

6. Please define what it is dross phase and how it is influencing on corrosion stability?

7. line 46. “Moreover, the density of bottom dross will be relatively smaller than zinc bath, with a decrease of”

The phrase is badly formulated. You need English editing for all article.

8. line 74. “show that Mg have the best corrosive resistance, lowest...”. It is wrong, Mg is very corrosion active metal. Only a small addition of Mg to alloy can be give corrosive protection.

9.  line 98.  “criteria esTablelished by”  mistake.

10. Figure 8. Caption, It is not corrosion rate, it is mass loss.

11. Figure 9, it is not clear why these figures needed (two completely identical bars) and the method of determination of the profile?.

12. It can be better to show on the cross section of the sample the distribution of the NaCl and cor. prod. It is not clear why the NaCl is at the surface (normally phases containing Cl- or Na+). You did not study the distribution of elements in cor. prod. in depth. 

Cl- gives mass gain and Na+ didn’t?

13. I do not think that this kind of layers are existing in reality.

14. line 243. “the sTableility of”  ?????

15. line 243-245 It is not clear.

16. The corrosion rate of  Al55Zn was high in presence of “Zn5(OH)8Cl2·H2O ” (the beginning slop on the Fig.8) and the corrosion rate decreased after 38 days of exposure due to formation of Zn0.67Al0.33(OH)2(CO3)0.165·xH2O . It is protective cor. prod.

17. line 247. “alloy soaking for different” replace to “ sample exposure for different”

18. line 252-254  “the reaction of electrochemical occurs”. We have many electrochemical reactions, please define, what do you mean.

19. line 363. “natural corrosion”  replace "corrosion during immersion in the aqueous electrolyte"

 20. line 368 Conclusion 2

21. “curve of 55AZS alloy increased steeply before 30 days, while increased gradually after 30 days is caused by a transition of corrosion product (Zn5(OH)8Cl2·H2O .Zn0.67Al0.33(OH)2(CO3)”

22. It is wrong. Reformulate.

23. Figure 8 shows that corrosion rate of 55 AZS (first slop on the curve) at the beginning is high and the rate decreased after 30 days of exposure. 24. Corrosion rate decreased (the second slop curve Fig.8) due to corrosion inhibition by formation of Zn0.67Al0.33(OH)2(CO3). After 30 days, the corrosion rate of the alloy 55AZS is very low and the corrosion rate (the second slop) was increased for Zn-Al-Mg.

24. In general, it can be important the influence of the dross particles on the corrosion rate. Them contain the Fe, which probably increased the electronic conductivity of dross phase that makes them more cathodic. The dross particles can participate in oxygen reduction and accelerate corrosion. You have to discuss this point in introduction and in the end of the article.